# Treatment of Radial Nerve Palsy in Paediatric Humeral Shaft Fractures—STROBE-Compliant Investigation

**DOI:** 10.3390/medicina58111571

**Published:** 2022-10-31

**Authors:** Łukasz Wiktor, Ryszard Tomaszewski

**Affiliations:** 1Department of Trauma and Orthopaedic Surgery, Upper Silesian Children’s Health Centre, 40-752 Katowice, Poland; 2Department of Trauma and Orthopedic Surgery, ZSM Hospital, 41-500 Chorzów, Poland; 3Institute of Biomedical Engineering, Faculty of Science and Technology, University of Silesia in Katowice, 40-007 Katowice, Poland

**Keywords:** humeral shaft fracture, radial nerve palsy, children

## Abstract

*Background and Objectives****:*** Due to the rarity of radial nerve palsy in humeral shaft fractures in the paediatric population and the lack of data in the literature, the purpose of our study was to report the treatment results of six children who sustained a radial nerve injury following a humeral shaft fracture. *Materials and Methods:* We treated six paediatric patients with radial nerve palsy caused by a humeral shaft fracture in our department from January 2011 to June 2022. The study group consisted of four boys and one girl aged 8.6 to 17.2 (average 13.6). The mean follow-up was 18.4 months. To present our results, we have used the STROBE protocol designed for retrospective observational studies. *Results:* We diagnosed two open and four closed humeral shaft fractures. Two simple transverse AO 12A3c; one simple oblique AO 12A2c; two simple spiral AO 12A1b/AO 12A1c and one intact wedge AO 12B2c were recognized. The humeral shaft was affected in the distal third five times and in the middle third one time. In our study group, we found two cases of neurotmesis; two entrapped nerves within the fracture; one stretched nerve over the bone fragments and one case of neuropraxia. We found restitution of the motor function in all cases. For all patients, extensor muscle strength was assessed on the grade M4 according to the BMRC scale (except for a patient with neuropraxia—M5). The differences in patients concerned the incomplete extension at the radiocarpal and metacarpophalangeal (MCP) joints. *Conclusions:* In our small case series, humeral shaft fractures complicated with radial nerve palsy are always challenging medical issues. In paediatric patients, we highly recommend an US examination where it is possible to be carried out to improve the system of decision making. Expectant observation with no nerve exploration is reasonable only in close fractures caused by low-energy trauma. Early surgical nerve exploration related with fracture stabilisation is highly recommended in fractures after high-energy trauma, especially in open fractures and where symptoms of nerve palsy appear at any stage of conservative treatment.

## 1. Introduction

Humeral shaft fractures are rare in the paediatric population with an overall prevalence of 0.4% to 3% of all paediatric fractures and 10% of all humerus fractures [1,2]. In comparison, humeral supracondylar fractures are one of the most common in children and they make up around 15% of all paediatric fractures [3]. Treatment of humerus diaphysis fractures is usually non-operative, except for open fractures, concomitant forearm fractures resulting in “floating elbow”, polytrauma cases or bilateral humeral fractures [1]. The radial nerve, due to its proximity to the bone, is at high risk of being damaged in humeral shaft fractures [4]. Radial nerve palsy is the most common nerve complication among long bone fractures and its prevalence in adults ranges from 7% to 17% [5,6,7,8,9,10]. There are considerable differences in opinion regarding the treatment of choice. The crucial question is whether to treat a radial nerve palsy conservatively or surgically, and if conservatively, at what stage should nerve exploration be considered. Studies based on adults have shown that such cases are always complex medical problems. In adults with symptoms of radial nerve damage in closed humeral fractures, in order to avoid unnecessary surgery policy, initial expectancy is recommended. Unfortunately, there is no clear recommendation on how long to wait before a surgical approach is taken. In open fractures with symptoms of radial nerve damage, early exploration should be considered. However, the literature on the above-mentioned problem in paediatric patients is limited, consisting mainly of individual case reports. We retrospectively evaluated the outcomes of the treatment of radial nerve palsy in a small series of children and adolescents treated at our department.

## 2. Materials and Methods

We have used the STROBE protocol designed for retrospective observational studies [11]. We treated 6 patients with radial nerve palsy in paediatric humeral shaft fractures between January 2011 and June 2022. An overview of the study group is presented in Table 1.

### 2.1. Case 1

An 8.6-year-old boy diagnosed with close, simple spiral fracture in the middle humeral thirds (AO 12A1b) after low-energy trauma. Radial nerve palsy occurred directly after initial humerus fracture. On the day of the injury, because of unacceptable fracture displacement, boy underwent close reduction with flexible intramedullary nailing without surgical nerve exploration. Radial nerve damage due to the closed nature of fracture and the low energy was qualified to expectant observation.

### 2.2. Case 2

A 16.3-year-old girl diagnosed with close, simple oblique fracture in the distal humeral thirds (AO 12A2c) after a high-energy trauma (suicide attempt, fall from 8 m). Among the accompanying injuries, we recognized: lungs contusion, bilateral pneumothorax, and fracture of the sacrum. The nerve damage had a primary nature. Due to a high-energy trauma, where a large part of the energy was focused on the arm, despite the fracture being closed, we selected it as a high risk of nerve damage. Since the patient’s condition was stable, early surgical nerve exploration with open reduction and internal plate fixation was performed on the day of the injury (neurotmesis was confirmed and patient needed nerve reconstruction with a sural nerve cable grafts).

### 2.3. Case 3

A 13.5 years-year-old boy diagnosed with close, simple transverse fracture in the distal humeral thirds (AO 12A3c) after low-energy trauma. Symptoms of the radial nerve damage did not appear at once after the injury, but 3 weeks later. In this patient, ultrasound (US) examination was performed and it helped to establish the indications for an operative nerve exploration. Open reduction with internal plate fixation and surgical nerve exploration was performed 23 days after the injury, less than 2 days after nerve palsy as a result of entrapment between the bone fragments.

### 2.4. Case 4

A 17.2-year-old boy diagnosed with open, intact wedge fracture in the distal humeral thirds (AO 12B2c; type 1 according to Gustilo Anderson classification) after high-energy trauma (fall from 6 m). The nerve damage had a primary nature. The patient sustained many additional injuries: right scapula fracture, multifragmentary fracture of the left distal forearm, stable Th8 compression fracture, multiple fractures of the pelvis, lungs contusion with a minor bilateral pneumothorax. Due to the bad patient condition, one day after initial trauma, we temporarily performed close reduction with external fixation and after next 15 days, we carried out an open reduction and internal plate fixation with surgical nerve exploration that revealed nerve entrapment between bone fragments.

### 2.5. Case 5

A 12.3-year-old boy diagnosed with open, simple transverse fracture in the distal humeral thirds (AO 12A3c; type 2 according to Gustilo Anderson classification) after a high-energy trauma (hit by a car). We diagnosed primary radial nerve palsy, and in addition: lung contusion, right distal radius fracture, numerous wounds, and bruises. On the day of the injury, patient underwent open reduction with flexible intramedullary nailing and no nerve exploration. Due to the deep contamination of the wound with mud and grass, radial nerve repair was postponed until the wound was healed and any possible infection had been limited (antibiotic prophylaxis was used). Electromyography carried out three months after the injury confirmed massive radial nerve damage at the humerus shaft level. After the wound had healed, and since nerve function had not returned, this patient underwent one-step implant removal and nerve reconstruction with a sural nerve cables graft 4.5 months after injury (nerve transection with no possibilities for end-to-end repair).

### 2.6. Case 6

A 17.5-year-old boy diagnosed with close, simple spiral fracture in the distal humeral thirds (AO 12A1c) after a low-energy trauma. In that patient, open reduction with internal plate fixation and nerve exploration was performed 56 days after the injury because of slowly progressive palsy due to the stretching over the bone fragments—subsequent radiograms showing increasing displacement of the bone fragments causing radial nerve stretching is shown in Figure 1. In this patient, due to the radial nerve palsy symptoms that started with a delay after the initial trauma, ultrasound (US) examination was performed. It helped to establish the indications for an operative nerve exploration. An intraoperative picture after radial nerve exploration and open reduction with internal fixation of the humeral shaft is shown in Figure 2. Such a proceeding resulted from the fact that patient was referred to our department from a distant hospital 6 weeks after initial trauma. Electromyography carried out outside our centre one month after the injury revealed massive radial nerve damage at the humerus shaft level.

During the radial nerve deficiency, the same scheme was used with initial application of a plaster cast and a subsequent orthosis supporting the wrist and hand. At the outpatient department, each patient had the control X-rays to assess progression of bone healing. After the fracture had healed, all patients remained under the rehabilitation clinic control, with the individual rehabilitation programme applied. Radial nerve recovery for each patient was assessed at subsequent follow-up visits. Muscular strength was evaluated using the British Medical Research Council (BMRC) rating scale [12]. This scale grades muscle power on a scale of 0 to 5 in relation to the maximum expected for that muscle—details in Table 2. The radial nerve is mainly a motor nerve, and therefore assessment of sensory deficits after nerve damage is less important than motor loss. Moreover, the radial nerve usually does not have an autonomic sensory zone (an area innervated by only one nerve); however, if it occurs, it is located between the I and II metacarpal bone above the first dorsal interosseous muscle, and there, the greatest sensation deficits could be found. Consequently, we abandoned the detailed classification of sensory disturbances, limiting assessment to detection of their presence.

All procedures in the study were in accordance with the ethical standards of the institutional and/or national research committee and with the 1964 Helsinki declaration and its later amendments or comparable ethical standards.

## 3. Results

The study group consisted of five boys and one girl aged 8.6 to 17.5 (average 14.23). The mean follow-up was 16.16 months (5 to 30). We diagnosed two open and four closed humeral shaft fractures: two simple transverse AO 12A3c; one simple oblique AO 12A2c; two simple spiral AO 12A1b/AO 12A1c and one intact wedge AO 12B2c were recognized. The humeral shaft was affected in the distal third five times and in the middle third one time. In our study group, we found two cases of neurotmesis; two entrapped nerves within the fracture; one stretched nerve over the bone fragments and one case of neuropraxia. We obtained bone healing in all six patients. Functional neurological outcomes were measured with grade and the time of recovery. Thumb, fingers, wrist extension deficits and muscular strength were evaluated using the BMRC rating scale. We found complete restitution of the motor function (grade M5 according to the BMRC scale for all extensor muscles) only in patient No. 1 in whom we initially diagnosed neuropraxia. Interestingly, for all other patients, extensor muscle strength was assessed on the grade M4 according to the BMRC scale, regardless of the type of nerve damage and time of surgical nerve exploration. The differences in those patients were the incomplete extension at the radiocarpal and metacarpophalangeal (MCP) joints. All patients had loss of extension of approximately 10 degrees at the radiocarpal joint (greater only for patient No. 5). Limitation of fingers extension at MCP joints of approximately 10 degrees was present for patients No. 4, 5 and 6. Similar extension deficit at the thumb MCP joint for patients No. 2, 5 and 6. Hypoaesthesia was found in half of the patients on the radio-dorsal part of the hand between I and II metacarpal bone (patients No. 2, 5 and 6). Individual extension range of motion, active muscular strength and sensory functions are presented in Table 3.

## 4. Discussion

Radial nerve palsy is the most common nerve complication among long bone fractures, and its prevalence in adults ranges from 7% to 17% [5,6,7,8,9,10]. A systematic review based on 21 papers (532 palsies in 4517 fractures) by Y.C. Shao et al. reported the prevalence of radial nerve palsy after fracture of the shaft of the humerus at 11.8% [6]. Some of these studies included paediatric patients; however, the number of children was very small and the exact contribution of children in the review was not specified. The frequency of radial nerve injuries in the paediatric population is significantly lower, but there are no reports in the literature. The vast majority of reports on the paediatric population are limited to the small case series. M.A. O’Shaughnessy et al. noticed four cases of radial nerve palsy caused by humeral shaft fracture out of 96 patients (4.2%); all those cases were neuropraxia [13]. Wrist drop, deficits in MCP joint extension and loss of a skin sensation are typical symptoms of radial nerve palsy; therefore, clinical examination is so crucial in making an accurate diagnosis [5,14,15,16,17]. All the muscles innervated by the radial nerve including triceps, brachioradialis, anconeus muscle, forearm supinator, wrist and finger extensors and abductor pollicis longus should be precisely tested. Clinical examination in children is less reliable and could be much more demanding for the physician, especially when it concerns young children with difficult cooperation. Plain radiographs allow the assessment of the level of humeral shaft fracture, and its morphology and displacement, therefore predicting the location and type of radial nerve damage.

### 4.1. Electrophysiological Studies

Electrophysiological studies are not reliable for identifying patients requiring radial nerve repair; furthermore, they are difficult to perform on children [10,18,19]. With this in mind, Bertelli, J. et al. propose that, in the paediatric population, surgery should be delayed for a minimum of 6 months to allow a longer time for spontaneous recovery and to avoid unnecessary surgical neurolysis [20]. Two patients in our study had EMG: patient No. 5, three months after the injury, and patient No. 6, one month after the injury. EMG confirmed massive radial nerve damage at the humerus shaft level in patient No. 5 (neurotmesis afterwards confirmed intraoperatively). In patient No. 6, EMG also showed massive radial nerve damage but with a few preserved motor fibres (inconsistently, anatomical continuity was preserved with confirmed intraoperatively nerve conduction). At the early stage after injury, EMG is not useful to distinguish between a damaged and disconnected radial nerve. Moreover, the cause of the nerve impairment cannot be easily assessed using electrophysiological tests. Therefore, in our opinion, the EMG has a limited value in the system of decision making.

### 4.2. Ultrasound Examination

US examination is helpful because it can detect nerve transection, nerve damage at the level of the hypoechogenic bone callus or nerve entrapment between bone fragments. It is non-invasive and accessible, but it requires a lot of experience from the examiner [5,14]. US is practically not applicable at the early stage after the fracture in connection with pathological humerus mobility and severe pain, especially in young and uncooperative children. Based on standards dedicated to adults proposed by Y.C. Shao et al., an US is recommended up to three weeks after the injury [6]. USs were performed in our study on two patients (No. 3, No. 6) and it helped both to establish the indications for the surgical nerve exploration. Our experience has shown that it is difficult to assess the longitudinal course of the nerve at the fracture level, which requires a lot of experience from the physician as well as patience from the patient. In our opinion, it is much easier to trace the radial nerve course in transverse US sections, and in the case of any abnormalities, complete the assessment of longitudinal sections. It needs to be highlighted that the radial nerve changed its shape from round at the proximal third of the humerus to oval at the middle third on the transverse US scan [21]. Ultrasound scans of the correct transverse section of the radial nerve are shown in Figure 3. In Figure 4, subsequent ultrasound scans of transverse radial nerve sections are shown (abnormal hypoechogenic nature of the radial nerve found at the level of stretching between the bone fragments). In our opinion, USs play a crucial role in the system of decision making.

### 4.3. Relationship between the Level of Fracture and the Radial Nerve Damage

Humerus diaphysis fractures located in the middle and distal third are related to a higher risk of radial nerve palsy because, at that level, the radial nerve passes through the intermuscular septum and becomes less mobile. The Holstein–Lewis fracture is a spiral distal third humeral shaft fracture typically related with radial nerve damage, which could be entrapped or lacerated by the displaced bone fragments [22,23]. Moreover, transverse (AO—12A3) and spiral (AO—12A1) fractures are more often associated with a nerve lesion compared to the other fracture types [3]. Our study confirmed the above statement; we recognized two simple transverse and one simple spiral fracture at the distal third of the humeral shaft (50% of all cases). As our cases have shown, in terms of nerve damage, there should be an increased awareness for humeral shaft fractures in the distal third not only directly after the initial trauma but also during follow-up after conservative treatment and even after surgical fracture stabilization.

### 4.4. Patient Management and Treatment Options

Based on the literature review, the main problem regarding the management of radial nerve palsy related with a humeral fracture among the paediatric group is the lack of approved standards or official consensus. In those particular cases, management usually relies on the recommendations established for adults. There is general agreement for radial nerve palsy that surgical exploration is accepted in case of: open fractures; fractures after high-energy trauma; when nerve palsy occurs after closed reduction with bone manipulation; and after penetrating injuries with relevant risk of nerve transection. Late surgical exploration should be performed when there is no nerve recovery after expectant observation. Due to a high rate of spontaneous nerve recovery, treatment is usually non-operative. There are inconsistent reports in the literature regarding the restitution of nerve function for different treatment approaches. There is also no agreement about the timing of the eventual surgery. Ekholm, R. et al. reported 71% of nerve recovery in adults after expectant observation with no surgical intervention, with a mean follow-up of 30 months (5.5–80) [23]. However, there is a lot of controversy regarding the expectant approach. A systematic review by Asif M. Ilyas et al. showed results that patients after expectant observation with no surgery present 77.2% of spontaneous nerve recovery, whereas patients after early (within 3 weeks) surgical exploration conjunct with fracture repair present 89.8% of recoveries. Patients with no improvement after expectant observation, subjected to late nerve exploration (after 8 weeks) have the worst results, estimated at 68.1% of recoveries [24]. An early surgical nerve exploration performed during the fracture repair could be easier than a delayed one; moreover, this could be safer for the patient because after fracture stabilization reduces the possibility of further nerve damage [6]. Based on this, and in tune with reports regarding the high prevalence of good results in our group, early nerve exploration was conducted on three patients: two of them presented nerve damage after high-energy trauma (patients No. 2 and No. 4), and one developed nerve palsy due to the fracture displacement over 3 weeks after initial trauma (patient No. 3). In our study, two patients underwent late nerve exploration: patient No. 5 after 4.5 months with the humeral fracture healed and patient No. 6 after 2 months with incomplete bone healing. In our survey, the results of a late surgical exploration of the radial nerve are good and correspond with the results of early exploration; this is most likely due to the greater nerve regeneration potential in children than in adults.

### 4.5. Surgical Techniques

In the case of radial nerve damage, many different techniques are proposed such as: direct end-to-end suture, nerve grafting if a nerve gap or segmental defect is present, and direct nerve transfer [5,8,25,26]. If the nerve cannot be repaired, tendons transfer can be a good option [3,5,27]. In our group, two patients required reconstruction with sural nerve cable grafts because end-to-end repair was impossible (No. 2 and No.5). In those cases, we obtained a good functional outcome corresponding to the report of J Bertelli et al. [20] who presented results of radial nerve reconstruction after distal humeral fractures. The remaining patients undergoing surgical exploration required nerve neurolysis and fracture fixation. The best final result was obtained in a patient with neuropraxia in whom we observed spontaneous recovery after 2.5 months of expectant observation.

### 4.6. Study Limitations

The main limitations of our study comprise the retrospective design and the small number of patients with humeral shaft fracture and concomitant radial nerve palsy.

## 5. Conclusions

In our small case series, humeral shaft fractures complicated with radial nerve palsy are always a challenging medical issue. In paediatric patients, we highly recommend an US examination where it is possible to be carried out to improve the system of decision making. Expectant observation with no nerve exploration is reasonable only in close fractures caused by low-energy trauma. Early surgical nerve exploration related with fracture stabilization is highly recommended in fractures after high-energy trauma, especially in open fractures and where symptoms of nerve palsy appear at any stage of conservative treatment.

## Figures and Tables

**Figure 1 medicina-58-01571-f001:**
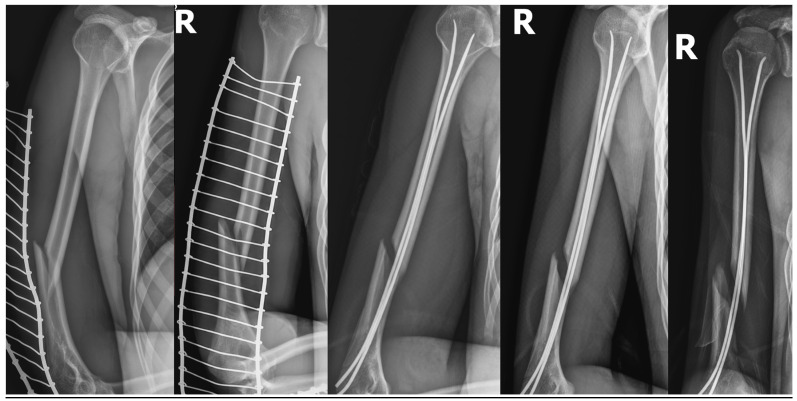
Patient No. 6. post-traumatic X-rays; subsequent control X-rays after close reduction and internal fixation/flexible intramedullary nailing (CRIF/FIN) showing increasing displacement of the bone fragments.

**Figure 2 medicina-58-01571-f002:**
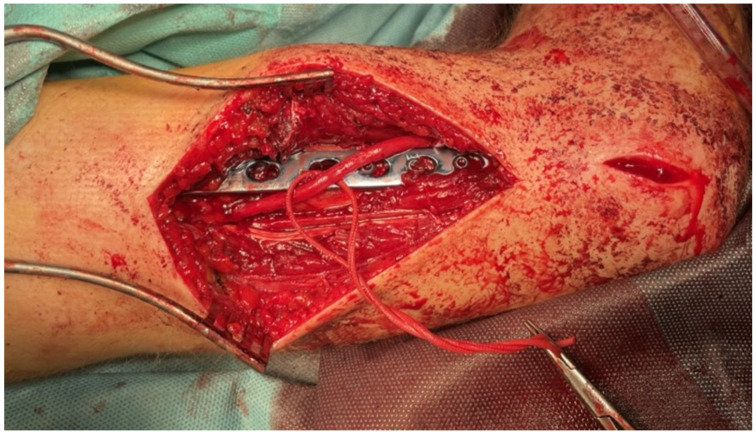
Patient No. 6. Intraoperative picture after nerve exploration and open reduction with internal fixation (ORIF). Intraoperative electrostimulation confirmed the preserved radial nerve continuity.

**Figure 3 medicina-58-01571-f003:**
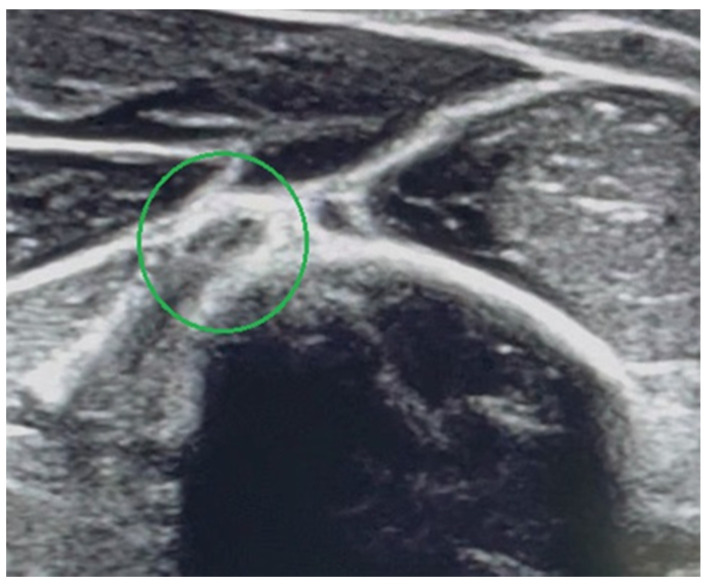
Ultrasound scan showing correct transverse section of radial nerve (the nerve and hyperechogenic outline of the humeral shaft are marked).

**Figure 4 medicina-58-01571-f004:**
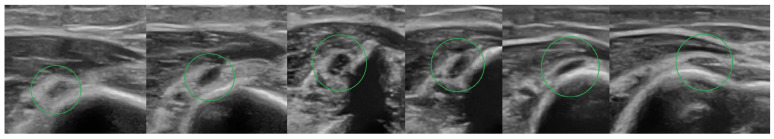
Patient No. 6. Subsequent ultrasound scans of transverse radial nerve sections. Abnormal hypoechogenic nature of the radial nerve found at the level of stretching between the bone fragments (the nerve and hyperechogenic outline of the humeral shaft are marked).

**Table 1 medicina-58-01571-t001:** Study group overview.

No.	Sex/Age	Fracture Type	Injury Mechanism	Accompanying Injuries	Type of Nerve Damage	Type of Intervention	Treatment Effect.	Follow -Up
1.	Boy/8.6 years	Close/ Simple spiral/Middle thirds AO 12A1b	Low-energy trauma		Neuropraxia	CRIF/FIN expectant observation	spontaneous recovery after 2.5 months	30 months
2.	Girl/16.3 years	Close/ Simple oblique/Distal thirds AO 12A2c	High-energy trauma/fall from 8 m	Lungs contusion. Bilateral pneumothorax. Fracture of the sacrum.	Neurotmesis	ORIF/titan plate + nerve reconstruction with a sural nerve cable graft.	recovery after 7.5 months	20 months
3.	Boy/13.5 years	Close/ Simple transverse/Distal thirds AO 12A3c	Low-energy trauma		Entrapment between bone fragments	ORIF/titan plate + nerve exploration 23 days after injury—2 days after nerve palsy.	recovery after 3.3 months	8 months
4.	Boy/17.2 years	Open GA 1/ Intact wedge/Distal thirds AO 12B2c	High-energy trauma/fall from 6 m	Right scapula fracture. Multifragmentary fracture of the left distal forearm. Stable Th8 compression fracture. Left lateral mass fracture of the sacrum. Right iliac wing fracture. Right pubic bone fracture. Lungs contusion with a minor bilateral pneumothorax.	Entrapment between bone fragments	CRIF/external fixator ORIF/titan plate + nerve exploration 16 days after injury	recovery after 4.6 months	16 months
5.	Boy/12.3 years	Open GA 2/Simple transverse/Distal thirds AO 12A3c	High-energy trauma/hit by a car	Numerous wounds and bruises. Lung contusion. Right distal radius fracture.	Neurotmesis	ORIF/FIN with no nerve exploration due to wound contamination; implant removal + nerve reconstruction with a sural nerve cable graft 4.5 months after injury.	recovery 5.5 months after nerve reconstruction/15.5 months after injury!	18 months
6.	Boy/17.5 years	Close/ Simple spiral/Distal thirds AO 12A1c	Low-energy trauma		Stretching over the bone fragments	CRIF/FIN ORIF/titan plate + nerve exploration 56 days after injury—slowly progressive palsy.	recovery after 2 months	5 months

**Table 2 medicina-58-01571-t002:** British Medical Research Council muscle power scale [12].

Grade	Muscle Power
0	no contraction
1	flicker or trace of contraction
2	active movement, with gravity eliminated
3	active movement, against gravity
4	active movement, against gravity and resistance
5	normal power

**Table 3 medicina-58-01571-t003:** Results of extension, active muscular hand strength and sensations disturbances.

No.	Wrist Extension	Fingers Extension	Thumb Extension	Active Wrist Extension Strength (BMRC)	Active Finger Extension (BMRC)	Active Thumb Extension (BMRC)	Skin Sensations Disturbances	Final Follow-Up (Months)
1.	No deficit	No deficit	No deficit	M5	M5	M5	No deficit	6
2.	10° deficit	No deficit	10° deficit/MCP joint	M4	M4	M4	Reduced skin sensation on radio-dorsal part of hand	9
3.	10° deficit	No deficit	No deficit	M4	M4	M4	No deficit	9
4.	10° deficit	10° deficit/MCP joint	No deficit	M4	M4	M4	No deficit	9
5.	15° deficit	10° deficit/MCP joint	10° deficit/MCP joint	M4	M4	M4	Reduced skin sensation on radio-dorsal part of hand	18
6.	10° deficit	10° deficit/MCP joint	10° deficit/MCP joint	M4	M4	M4	Reduced skin sensation on radio-dorsal part of hand	5

## Data Availability

The data generated and analysed in the current study are available from the corresponding author on reasonable request.

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
