# Peer review of "Treatment of Radial Nerve Palsy in Paediatric Humeral Shaft Fractures—STROBE-Compliant Investigation"

_medicina, 2022, doi:10.3390/medicina58111571_

Round 1
Reviewer 1 Report
It's un interesting paper that confirm what we already know.
The main question addressed by the research is when we have a radial nerve palsy, after humeral fracture, even if rare, what we should do: to wait and see?; to proceed for surgery, immediately?
This topic is quite original.
Compared with other published material, not so much is add to the subject but the paper give some protocol which would be useful. The paper is well written and text clear and easy to read. The conclusions are consistent with the evidence and arguments presented.They address the main question posed.
Author Response
Dear reviewer.
We would like to kindly thank you for your time spent reviewing our manuscript ‘’ Treatment of radial nerve palsy in paediatric humeral shaft fractures. STROBE-Compliant Investigation’’. We appreciate all your valuable comments of our work. We would like to emphasize that all revisions made were marked up using the “Track Changes” function in MS Word so changes can be easily viewed.
Best regards.
Łukasz Wiktor
Reviewer 2 Report
The article describes a case series of 6 pediatric patients with radial nerve palsy following humeral fracture.
The topic is interesting; the management of post-trauma radial nerve palsies is still debated even in adults, even more so in children, considering the rarity of the cases.
Considering this, I think this article can make a contribution to the literature. The article is quite well written but some major revisions are needed in my opinion. In particular:
1) considering the wide variability of treatments on the patients described, I think describing patient-by-patient clinical setting and management individually is more functional to make the cases clearer for readers
2) early exploration was not conducted for case #5, despite the fact that there is a clear indication in adults to do so in case of open fractures; please better justify this choice
3) timing of treatments is crucial; please state more precisely the timing of surgical treatment, of any early exploration or late explorations, as well as the timing of nerve reconstructions when performed
4) the table regarding neurological functional recovery is interesting, but please also include the timing of evaluation (last follow-up?)
Finally, I congratulate the authors for the exhaustive Discussion. I think the introduction needs to be lengthened with some information regarding the management of radial palsy in the adult.
Thank you.
Author Response
We would like to kindly thank you for your time spent reviewing our manuscript ‘’ Treatment of radial nerve palsy in paediatric humeral shaft fractures. STROBE-Compliant Investigation’’. We appreciate all your valuable comments of our work. We have revised our manuscript, according to your suggestions. We believe that the manuscript has been further improved. All revisions made were marked up using the “Track Changes” function in MS Word so changes can be easily viewed.
Best regards.
Łukasz Wiktor
1) considering the wide variability of treatments on the patients described, I think describing patient-by-patient clinical setting and management individually is more functional to make the cases clearer for readers.
We have completely changed the section of material and methods. We have introduced the description patient-by-patient as you recommended.
2) early exploration was not conducted for case #5, despite the fact that there is a clear indication in adults to do so in case of open fractures; please better justify this choice.
We have added a justification in the material and methods section. “On the day of the injury patient underwent open reduction with flexible intramedullary nailing and no nerve exploration. Due to the deep contamination of the wound with mud and grass, radial nerve repair was postponed until the wound is healed and any possible infection is limited (antibiotic prophylaxis was used). Electromyography carried out three months after the injury confirmed massive radial nerve damage at humerus shaft level. After the wound has healed, since nerve function hasn't returned this patient underwent one-step implant removal and nerve reconstruction with a sural nerve cables graft 4.5 months after injury (nerve transection with no possibilities for end-to-end repair)”.
3) timing of treatments is crucial; please state more precisely the timing of surgical treatment, of any early exploration or late explorations, as well as the timing of nerve reconstructions when performed.
We have completed the missing data.
4) the table regarding neurological functional recovery is interesting, but please also include the timing of evaluation (last follow-up?).
We have added to table 3 a column including the timing of evaluation.
Finally, I congratulate the authors for the exhaustive Discussion. I think the introduction needs to be lengthened with some information regarding the management of radial palsy in the adult.
We have added some information regarding the management of radial palsy in the adult to the introduction section as you recommended. “The crucial question is whether to treat a radial nerve palsy conservatively or surgically, and if conservatively, at what stage the nerve exploration should be considered. Studies based on adults have shown that such cases are always a complex medical problem. In adults with symptoms of radial nerve damage in closed humeral fractures in order to avoid unnecessary surgery policy of initial expectancy is recommended. Unfortunately, there is no clear recommendation how long to wait before surgical approach. In open fractures with symptoms of radial nerve damage, early exploration should be considered”.
Round 2
Reviewer 2 Report
The Authors have addressed all my concerns. Thank you.